**The dynamical role of upper layer salinity in the Mediterranean Sea**
Ali Aydogdu[1], Pietro Miraglio[1], Romain Escudier[2], Emanuela Clementi[1], Simona Masina[1]
[1] Ocean Modeling and Data Assimilation Division, Fondazione Centro Euro-Mediterraneo sui Cambiamenti Climatici,
Bologna, Italy
[2] Observations pour Les Systèmes D'analyse et de Prévision, Mercator Ocean International, Toulouse, France
*Correspondence to*: Ali Aydogdu (ali.aydogdu@cmcc.it)
**Abstract.**
The Mediterranean Sea is a semi-enclosed basin with an excess amount of evaporation compared to the water in-flux through
precipitation at the surface and river runoff on the land boundaries. The deficit in the water budget is balanced by the inflow
in the Gibraltar Strait and Turkish Straits System connecting the Mediterranean with the less saline Atlantic Ocean and the
Black Sea, respectively. There is evidence that the Mediterranean region is a hotspot in a warming climate which will possibly
change the water cycle significantly, but with large uncertainties. Therefore, it is inevitable to monitor the evolution of the
essential ocean variables to respond to the associated risks and mitigate the related problems. In this work, we evaluate the
evolution of the salinity content and anomaly between 0-300 m in the Mediterranean Sea during the last decades using the
Copernicus Marine Service reanalysis and in-situ objective analysis products. The results show an increasing mean salinity
with a stronger trend in the eastern basin. The spread of the products implies a larger variability in the western basin while
the standard deviation is lower in the eastern side, especially in the Ionian and the Levantine basins.

**Short summary.**
This paper investigates the salinity content and anomaly evolution in the Mediterranean Sea using observational and
reanalysis products. The salinity content increases overall while negative salinity anomalies appear in the western basin
especially around the upwelling regions. There is a large spread in the salinity estimates that reduces with the emergence of
the Argo era.
**1 Introduction**
The Mediterranean Sea is warming (Pisano et al., 2020). It is evaporating more and more (Skliris et al., 2018; Jordà et al.,
2017) with marine heat waves increasing in intensity, duration and frequency (Juza et al., 2022; Dayan et al., 2022). The
Mediterranean region is a hotspot with global warming (Tuel and El Tahir, 2020) that will likely alter the water cycle (Cos
et al., 2022). Tracking the changes of the essential ocean variables (EOVs) is crucial in order to understand the impact of
climate change. Two of these EOVs are linked to the ocean salinity at the surface and subsurface, which will be affected
significantly by the surface heat and freshwater fluxes. The global water cycle modulating the ocean salinity is a key element
of the Earth's climate (Cheng et al., 2020). In the Mediterranean Sea, freshwater fluxes through the land (rivers) and
atmosphere (evaporation and precipitation) are balanced by two sea straits, namely Gibraltar and Dardanelles, from which
the less saline Atlantic Ocean and Black Sea waters flow into the basin with an annual net inflow of $0.78 \pm 0.05$ Sv (Soto-
Navarro et al., 2010) and $0.05 \pm 0.04$ Sv (Jarosz et al., 2013), respectively. These density contrasts contribute to the wind
driven circulation and generate a highly energetic anti-estuarine circulation (Cessi et al., 2014). The salinity of the Atlantic
water entering through the Gibraltar Strait is about 36.2 psu. The salinity of the Dardanelles can vary significantly and it can
be as low as 27 psu (Aydogdu et al., 2018; Sannino et al., 2017). Recently, Fedele et al. (2022) studied the characterization
of the Atlantic Waters (AW) and Levantine intermediate waters (LIW) from the ARGO profiles in the last 20 years. Their
conclusion is a clear salinification and warming trend which characterised both AW and LIW over the last two decades.
Skliris et al. (2018) argue that the Mediterranean basin salinification is driven by changes in the regional water cycle rather
than by changes in salt transports at the straits, as it is shown by the water mass transformation distribution in salinity
coordinates. However, we will show that there is a bigger uncertainty compared to most of the basin in the radius of influence
of both the Gibraltar and Dardanelles Straits. In Section 2, the data and methods used in this study are presented. In Section
3, the results are shown and discussed, while in Section 4 the conclusions are drawn.
**2 Data and method**
In this study, Copernicus Marine Service global and regional reanalysis as well as observational gridded products are used to
explore the role of the salinity variability in the 0-300 m depth among different estimates as well as temporal and spatial
anomalies against a mean.
The Mediterranean 1/24° resolution regional reanalysis (hereinafter, MEDREA24; Escudier et al., 2021) from Copernicus
Marine Service is used as a regional high-resolution product. In this work, MEDREA24 and its interim extension until the
end of 2021 are included. Moreover, the 1/4° resolution Global Reanalysis Ensemble Product (hereinafter, GREP) is also
used. It consists of the global reanalysis from Mercator Ocean's GLORYS2V4 (Lellouche et al., 2013), UK MetOffice's
GLOSEA5v13 (MacLachlan et al., 2014) using the FOAM system (Blockley et al., 2014), CMCC C-GLORSv7 (Storto et
al., 2016) and ECMWF's ORAS5 (Zuo et al., 2017). A study on the ocean heat content and steric sea level representation in
the GREP ensemble can be found in Storto et al. (2019a). A more general status of the global ocean reanalysis is reviewed
in Storto et al. (2019b). The period covered by the GREP is 1993-2019.
As observational products, the CORA (Szekely et al., 2019) and ARMOR3D (Guinehut et al., 2012) gridded reconstructions
are adopted. Both datasets are available between 1993 and 2020. In the CORA, the objective analysis is performed on
measurement's anomalies relative to a first guess provided by World Ocean Atlas 2013 monthly climatology for different

decades, to accurately reproduce the climate tendencies, is interpolated and centred on the 15th of each month. Instead in ARMOR3D, the first guess is adopted from World Ocean Atlas 2018. Both products use an objective analysis method proposed by Bretherton et al., (1976).

The investigation is performed in the entire Mediterranean Sea as well as in the eastern and western basins, which have very different characteristics. The salinity mean ($\overline{S}$) is computed using formula (1) as the monthly volume ($V$) average of each product between 0 and 300 m depth, i.e., excluding the shelf areas close to the coast with a depth less than 300 m.

$$\overline{S} = \frac{1}{V} \int_V S\, dV$$

(1)

The mean of different products and their standard deviation are evaluated in the common period 1993-2019. Besides this time frame, the CORA and ARMOR3D time series are available until 2020, while the MEDREA24 time series is provided up to 2021 (last 6 months extended in interim mode).

The salinity anomalies are computed using formula (2). The difference of the salinity, $S$, with a reference salinity, $S_{ref}$, is normalised by the depth of the water column which is constant in our case with $z_2 = 300$ m and $z_1 = 0$ m.

$$S_A = \int_{z_1}^{z_2} \frac{S - S_{ref}}{z_2 - z_1} dz$$

(2)

Formula (2) is a modified version of the one proposed in Boyer et al. (2007) which uses the salinity as a proxy for the equivalent freshwater content. This method has been later adopted in various studies including, among the others, Holliday et al. (2020), with a density weight to account for baroclinic properties of the water column. The formulation in Boyer et al. (2007) is based on a reference salinity. As an example, mean values of a basin (Aagaard and Carmack, 1989) is a widely used choice for $S_{ref}$ in global freshwater content calculations. However, it is argued that since a reference value can be chosen arbitrarily, this would bring ambiguity (Schauer and Losch, 2019) in computing the equivalent freshwater content. Therefore, in this study we propose to evaluate the salinity content and anomaly following formula (2) by choosing $S_{ref}$ as a monthly climatology of each dataset computed from each product separately between 1993 and 2014. This period is chosen to be consistent with the Ocean Monitoring Indicators produced previously in the Mediterranean Sea and other Copernicus Marine domains. Furthermore, the calculations are performed in the entire Mediterranean Sea (MED) as well as in its western (WMED) and eastern (EMED) sub basins, separated at the Sicily Strait. In Fig. 1, we present the monthly variation of the $S_{ref}$, as an example only the one from MEDREA24, which shows a clear difference in the seasonality in the EMED and WMED, with a maximum in March and December, respectively. It is also evident that the Mediterranean monthly salinity

reference shows a seasonal cycle much similar to the one of the Eastern basin (but with different magnitude) characterised
by lower salinity during the summer period and larger values at the end of the year.

**3 Results and Discussion**

The Copernicus Marine service products described in the previous section allow the assessment of the salinity content of the
Mediterranean Sea along with its anomaly and trend during the last decades.
In Fig. 2, we present the time series of the mean salinity content in the first 300 m derived from the analysed products
(MEDREA24 in red; GREP ensemble mean in blue; GREP ensemble members in thin light blue; CORA in dark green,
ARMOR3D in light green) and their overall mean (in black) and spread (shaded grey).
In the early 90s in the entire Mediterranean Sea (Fig. 2a), there is a large spread in salinity with the observations showing a
higher salinity while the reanalysis products present relatively lower salinity. This is the case until 2005. Coinciding with the
global coverage of the Argo profilers in the early 2000s following the efforts in the Global Ocean Data Assimilation
(GODAE) together with the Climate Variability and Predictability Programme (CLIVAR) and the Global Climate Observing
System/Global Ocean Observing System (GCOS/GOOS), the spread among different products narrows. Possibly, the
reanalyses are better constrained through data assimilation with this novel observation type (Johnson et al., 2022) which
provides high-resolution and high-frequency temperature and salinity profiles all over the world' ocean while the observation-
based gridded products become more confident. The maximum spread between the period 1993-2019 is in the 90's with a
value of 0.096 psu and it decreases to as low as 0.009 psu by the end of 2010s. The mean salinity computed in the entire
Mediterranean Sea from all products varies between approximately 38.5 and 38.6 psu with a spatiotemporal mean of 38.57
psu (Table 1).
In the western Mediterranean (Fig. 2b), the overall mean is centred around 38.16 psu with a larger spread - with a maximum
and minimum of 0.172 psu and 0.026, respectively - occurring in the early 2000s. An increase of the mean salinity in 2005
is evident from all the reanalysis products and, at a lesser extent, from the CORA dataset, for which one of the many possible
reasons is the regime shift as discussed in (Schroeder et al., 2016) corresponding to a major deep water formation event at
the beginning of the Western Mediterranean Transition (Zunino et al., 2012).
In the eastern Mediterranean (Fig. 2c), the overall mean is centred around 38.87 psu with a lower spread compared to the
western basin with a maximum and minimum of 0.086 psu and 0.003, respectively.
Overall, for the period between 1993-2019 we note that the observational products, gridded using optimal interpolation
statistical techniques, show a higher average salinity compared to the reanalysis products that are dynamically integrated and
corrected through data assimilation. The spread is representing the offset of the products more than their variability in the
entire Mediterranean Sea, as well as in its eastern and western subdomains.

All the products show a positive trend between 1993-2019 (in parenthesis in Table 1). The trend in the mean of all products is calculated as 0.0056 psu/year. This trend is consistent with the estimates between 1950-2002 of Skliris et al., (2018) from EN4 and MEDATLAS data sets which shows a trend of $0.0096 \pm 0.0077$ and $0.0088 \pm 0.0092$ respectively, in the first 150 m while $0.0067 \pm 0.0040$ and $0.0067 \pm 0.0036$ between 150-600 m. The differences in trend in different products that we used are mainly due to the discrepancies at the beginning of the timeseries. The weak consistency among the reanalyses visible during the first decade is likely due to the lack of observations not sufficient to constrain the different models which use different physics and initialization (e.g., Masina and Storto, 2017). The reduction of the spread among the products evolves in parallel to the increase of the observational coverage after the advent of the ARGO network. The observational products will be impacted from the scarcity of the observations in the '90s, since they rely on statistical methods. The trend calculated for each grid point from the MEDREA24, which is the analysed product covering the longest period, is presented in Fig. 2d. The dominant signal in the entire basin is positive with a larger amplitude in the Balearic Sea, Ionian Sea, Adriatic Sea, Western Levantine and with a less evident signal in the Gulf of Lions, Northern Aegean Sea and Eastern Levantine Basin. A small negative trend zone appears in the Alboran Sea. The trend in the entire analysed period is about 0.0049 psu/year in the western basin. This is below the rate of the basin-wide trend which is larger due to the trend in the eastern basin (0.0061 psu/yr). Differences among different products, especially the objectively-analysed observations and GREP, for the trend is larger in the western basin. They are more confined around the mean in the eastern basin which may explain also the lower spread in this area as discussed below in Fig. (3b).

For 2020, CORA and ARMOR3D products are available, and both continue to sustain the positive trend even though it is less evident in the western basin. MEDREA24 (and its interim extension) shows an increasing mean salinity until the end of 2021. All products present larger values after 2016 and a maximum in 2018.

The spatial mean, computed between 1993-2014 from all products in the first 300 m (Fig. 3a), shows a gradual increase in the upper ocean integrated salinity from west to east. Minimum salinity occurs close to river mouths, such as in the North Adriatic Sea due to the freshwater input from the Po River, and on the pathways of the outflow of the Dardanelles and Gibraltar straits. The Atlantic water, modified through its route, can be traced till the eastern basin from its low salinity. The spread deduced from all the products (Fig. 3b) implies that they agree more, meaning lower spread, in the Levantine and Ionian Seas and to a lesser degree between the Balearic and Sardinia / Corsica islands. The spread is larger especially in the northern Aegean and Adriatic Sea and southwestern coast between the Gulf of Gabes and Gibraltar Strait. This uncertainty or mismatch in the products is possibly due to the different volume fluxes through the rivers and straits.

In Fig. 4 (a-c), we show the time series of the salinity anomaly estimates in the western (Fig. 4a), eastern (Fig. 4b) and entire (Fig. 4c) basin from each product using formula (2). We recall that the salinity reference is computed for each product per se. Moreover, the salinity anomaly map in 2021 from MEDREA24 is depicted in Fig. (4d), computed against the overall mean between 1993-2014, which is shown in Fig. (3a).

The anomalies have a larger range in the reanalysis products. There is a negative anomaly within the first decade in GREP
and MEDREA24 which turns to positive first in the western Mediterranean (Fig. 4a) and followed by the eastern basin (Fig.
4b) after 2006. In the CORA and ARMOR3D, instead, there is a clear increase in the salinity anomaly in the eastern
Mediterranean and the entire basin with a less evident positive trend in the western basin. We summarise the mean salinity
anomalies in Table 2.
In 2021, the anomaly is mostly positive with some negative anomaly structures on the path of Atlantic water (Fig. 4d),
Alboran Sea, upwelling favouring Balearic Islands. Fedele et al. (2022) reports a positive salinity trend in the modified
Atlantic and Levantine intermediate waters using 18-year-long (2001-2019) Argo profiles, which in general agrees with the
anomaly map to a large extent. However, we note that the spread on the pathway of the water entering from the Gibraltar
strait and reaching the Levantine basin has a relatively larger spread compared to the deeper areas (see Fig. 3b).
**4 Conclusions**
In this study, we presented the salinity characteristics of the Mediterranean Sea in the upper 300 m deduced from various
products including reanalysis and gridded observational datasets released by Copernicus Marine Service. The products with
dynamically constructed ocean reanalysis and objectively analysed observations show significantly large spread at the
beginning of the period of investigation while the uncertainty reduces possibly with the emergence of ARGO profilers which
allowed a wider spatial and higher frequency sampling in the ocean. The mean salinity with its anomaly and trend is computed
and analysed in the entire basin as well as in the western and eastern basins for all the datasets separately and averaged. The
spatial maps of the mean and the spread of the salinity are depicted and discussed. The overall results show a salinification
of the Mediterranean Sea agreeing with earlier studies (e.g., Skliris et al. 2018). The subbasin scale investigation shows
negative salinity anomalies in the western basin in the upwelling regions, which may imply stronger upwelling events, and
waterway following the north African coast, which may be a consequence of the freshening North Atlantic water masses
(Holliday et al., 2020). There is a large spread in the salinity estimates among different products, which reduces with the
introduction of the Argo profilers in the data assimilation components of the reanalysis systems. Besides the large spread,
considering the reported discrepancies in the salinity measurements after 2016 (Barnoud et al., 2021), it is essential to use all
available information sources for a more accurate state estimate and uncertainty quantification.
**Data availability**
All datasets used in this article can be obtained from the Copernicus Marine Service catalogues as described in Table 3 with
their names, temporal coverages, and documentation.

**Acknowledgments**

This study has been conducted using EU Copernicus Marine Service Information. This work has been funded through the EU Copernicus Marine Med-MFC Service Lot n. 21002L5-COP-MFC MED-5500.

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

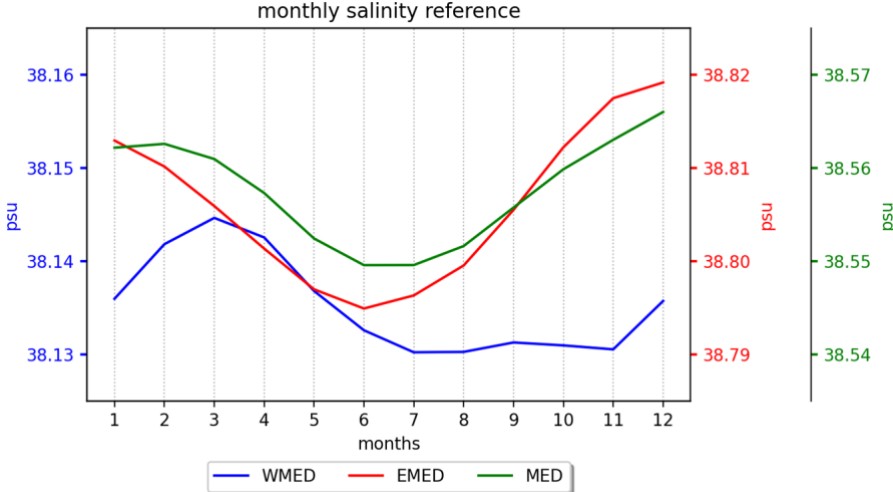

**Figure 1:** The monthly reference salinity $S_{ref}$ estimates calculated from the MEDREA24 in the period 1993-2014. The green, blue, and red curves show the MED, WMED and EMED regions respectively on its corresponding vertical axis. The same calculation is done for each product separately (not shown) to evaluate formula (2) to compute salinity anomaly.

| psu (psu/year) | MED | WMED | EMED |
|---|---|---|---|
| MEDREA24 | 38.58 (0.0070) | 38.15 (0.0055) | 38.83 (0.0078) |
| GREP | 38.48 (0.0110) | 38.01 (0.0111) | 38.87(0.0106) |
| CORA | 38.61 (0.0020) | 38.21(0.0019) | 38.84 (0.0024) |
| ARMOR3D | 38.64 (0.0027) | 38.24 (0.0020) | 38.75(0.0032) |
| mean | 38.57 (0.0056) | 38.16 (0.0049) | 38.87 (0.0061) |

**Table 1.** The temporal mean salinity (in psu) and trend (in psu per year) in the 0-300 m between the common period 1993-2019 for separate products and their overall mean in *Fig. 2*.

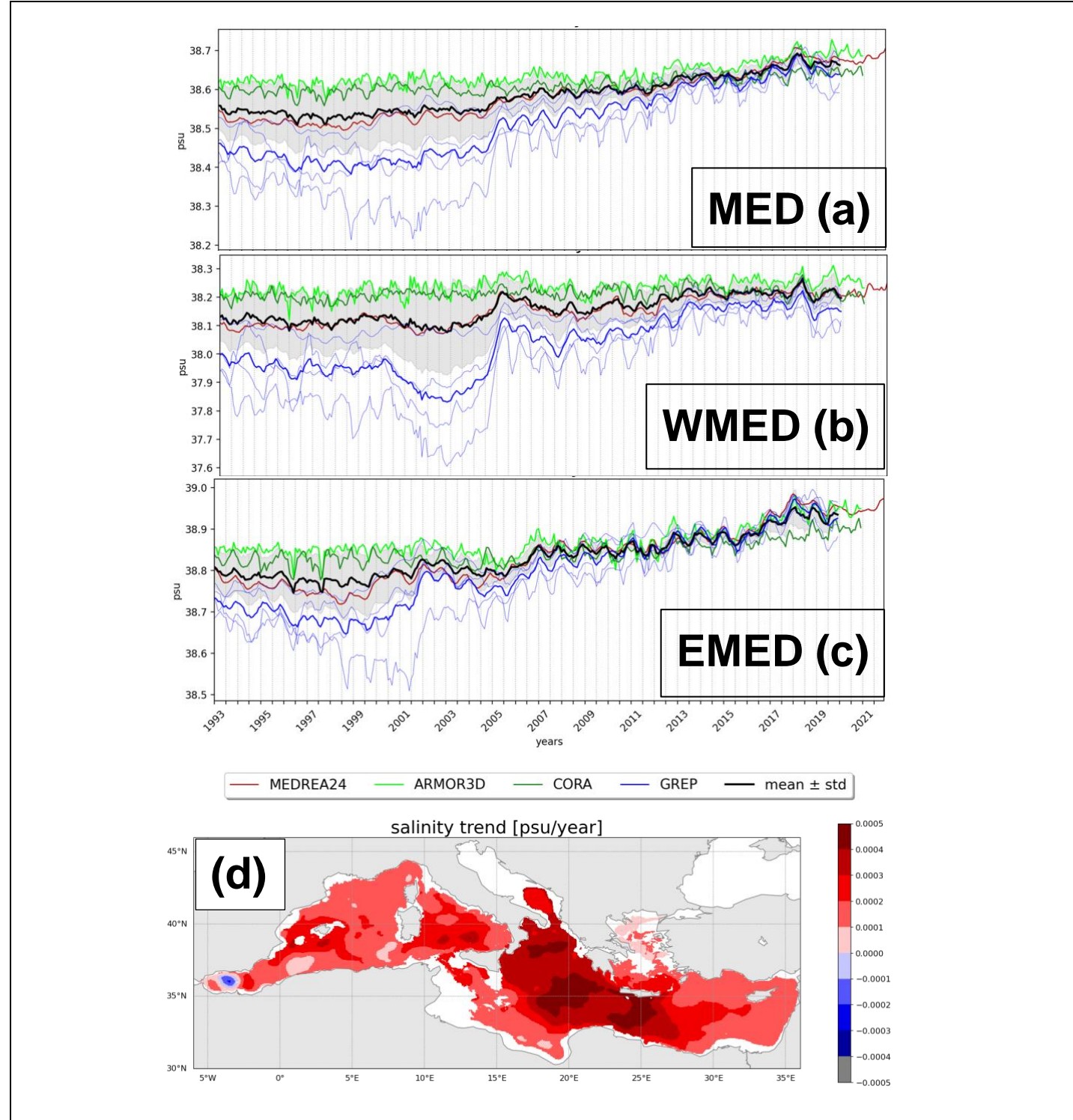

272

**Figure 2:** Time series of mean salinity in the upper 300 m in the **(a)** entire Mediterranean Sea **(b)** western Mediterranean basin and **(c)** eastern Mediterranean basin between the period 1993 and 2021 from MEDREA24, until the end of 2020 for ARMOR3D, CORA and until the end of 2019 for GREP. The mean of all the products is drawn in black with their standard deviation shaded around the mean. GREP ensemble members are depicted in thin blue curves. The GREP product covers the period until 2019 while the observational products CORA and ARMOR3D cover until 2020. The time series for MEDREA24 is extended until 2021 using the interim products. (d) salinity trend in the first 300 m. from the MEDREA24 psu per year.

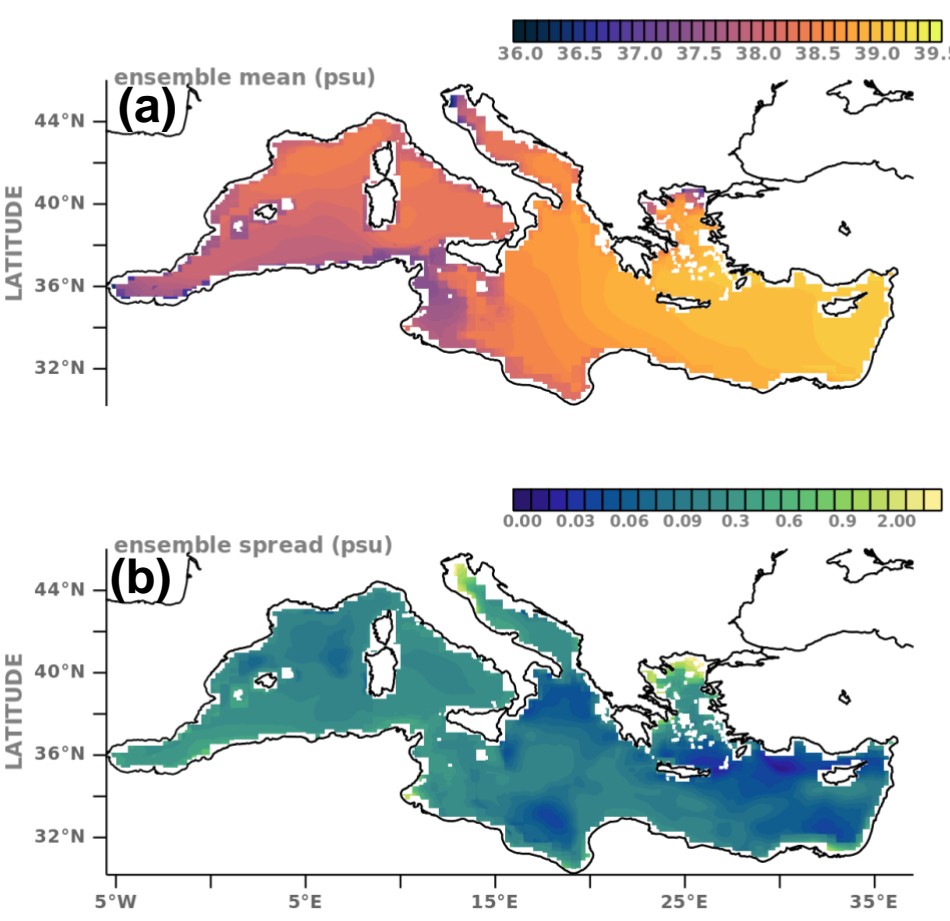

**Figure 3.** The maps of **(a)** mean and **(b)** spread of integrated salinity in the period between 1993–2014 in 0-300 m. computed from the GREP ensemble mean, CORA, ARMOR3D and MEDREA24 products. We refer to the text for the information on the data products used. The analysis is performed only if the water column is deeper than 300 m. Note that in **(b)** the colour scale is not linear to show the smaller standard deviation.

| psu | MED | WMED | EMED |
|---|---|---|---|
| MEDREA24 | 0.026 | 0.027 | 0.025 |
| GREP | 0.042 | 0.056 | 0.034 |
| CORA | 0.001 | -0.008 | 0.007 |
| ARMOR3D | 0.001 | -0.009 | 0.008 |
| mean | 0.018 | 0.017 | 0.019 |

**Table 2.** The temporal mean salinity anomaly (in psu) in the 0-300 m between the common period 1993-2019 for separate products and their average in *Fig. 3*.

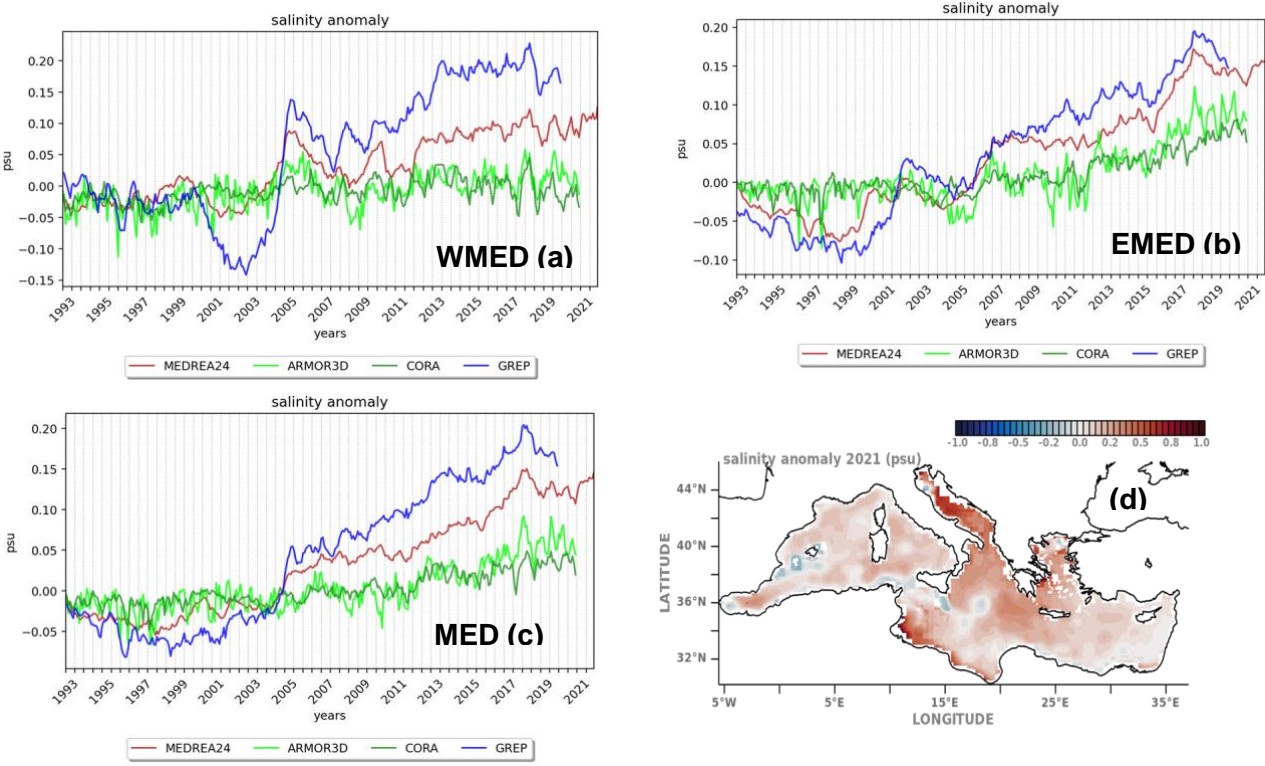

288

**Figure 4.** Time series of salinity anomaly from the MEDREA24, GREP, ARMOR3D and CORA products in the **(a)** western

Mediterranean Sea **(b)** eastern Mediterranean basin and **(c)** entire Mediterranean basin computed with respect to the monthly

reference salinity estimates in the corresponding area in **Fig. 1** calculated from the MEDREA24 in the period 1993-2014.

The GREP products cover the period until 2019 while the observational products CORA and ARMOR3D covers until 2020.

The time series for MEDREA24 is extended until 2021 using the interim products. **(d)** salinity anomaly in 2021 in the

Mediterranean Sea against the mean of salinity in Fig. 3a.

|  | Product Name | Documentation | Data access / Time Period |
|---|---|---|---|
| Mediterranean Sea Physics Reanalysis | MEDSEA_MULTIYEAR_PHY_006_004; Numerical models | Product User Manual (CMEMS-MED-PUM-006-004)<br>Quality Information Document (CMEMS-MED-QUID-006-004)<br>https://resources.marine.copernicus.eu/product-detail/MEDSEA_MULTIYEAR_PHY_006_004/DOCUMENTATION<br><br>Escudier et al., (2020) | EU Copernicus Marine Service Product 2022 / 1987-2021 |

| | | | |
|---|---|---|---|
| Global Ocean Ensemble Physics Reanalysis | GLOBAL_REANALYSIS_PHY_001_031; Numerical models | Product User Manual (CMEMS-GLO-PUM-001-031)<br>Quality Information Document (CMEMS-GLO-QUID-001-031)<br>https://resources.marine.copernicus.eu/product-detail/GLOBAL_REANALYSIS_PHY_001_031/DOCUMENTATION | EU Copernicus Marine Service Product 2022 / 1993-2019 |
| Global Ocean-Delayed Mode gridded CORA-In-situ Observations objective analysis in Delayed Mode | INSITU_GLO_TS_OA_REP_OBSERVATIONS_013_002_b; Observations | Product User Manual (CMEMS-INS-PUM-013-002-ab)<br>Quality Information Document (CMEMS-INS-QUID-013-002b)<br>https://resources.marine.copernicus.eu/product-detail/INSITU_GLO_TS_OA_REP_OBSERVATIONS_013_002_b/DOCUMENTATION | EU Copernicus Marine Service Product 2022 / 1993-2020 |
| Multi Observation Global Ocean 3D Temperature Salinity Height Geostrophic Current and MLD | MULTIOBS_GLO_PHY_TSUV_3D_MYNRT_015_012; Observations | Product User Manual (CMEMS-MOB-PUM-015-012)<br>Quality Information Document (CMEMS-MOB-QUID-015-012)<br>https://resources.marine.copernicus.eu/product-detail/MULTIOBS_GLO_PHY_TSUV_3D_MYNRT_015_012/DOCUMENTATION<br><br>Guinehut et al., (2012) | EU Copernicus Marine Service Product 2022 / 1993-2020 |

295 **Table 3.** Products from Copernicus Marine Service used in this study.