# Peer review of "The dynamical role of upper layer salinity in the Mediterranean Sea"

_State of the Planet, 2022_

## Author Comment (AC1)

**Response to the Reviewer #1 for sp-2022-11**

We thank the reviewer for the constructive feedback on our manuscript. Below are our point by point responses to the issues raised on the general aspects and minor suggestions. The reviewer comments are highlighted with italic typeset with grey fonts while our response is in default typeset.

*This short manuscript describes the evolution of upper ocean salinity content in the Mediterranean Sea by using state-of-the-art global and regional reanalyses and objective analyses. As such, it is of interest for the Mediterranean climate community. It is concise, well written, and easy to read. There are, however, a few aspects that need to be improved before the manuscript is suitable for publication. Given that all the points below are relatively easy to address, I recommend a minor revision.*

Thank you.

**General points**

*1) Not clear why the reference S_ref is calculated over 1993-2014 and not the full period. By doing this, the salinity anomaly figure / diagnostics (as the mean in Table 2) basically represent the difference between <1993-2021> minus <1993-2014> so the anomaly of the latest 7 years, which is somehow subjective and not necessarily easy to interpret. I would expect S_ref to be the full period, otherwise it needs to be justified with some arguments.*

Thanks for the opportunity to clarify this issue. We completely understand the concern of the reviewer. We would like to emphasise that this study is a contribution to the Ocean State Report (OSR) 7 which will be proposed as an Ocean Monitoring Index (OMI) if it is published. The period to calculate the mean is chosen to be consistent with the previous contributions to OSR and current OMIs provided in the Mediterranean Sea, from the beginning of the OSR . Since the salt content anomaly OMI will be updated every year, as in example as in Fig. 4d, we would like to keep the mean that is used to calculate anomalies fixed. We note that the longer timeseries come from MEDREA24 and observational products, while GREP products are now available only to the end of 2019  even if they are regularly extended. With the hope that it clarifies our reasoning, we added a sentence as "This period is chosen to be consistent with the Ocean Monitoring Indicators produced previously in the Mediterranean Sea and other Copernicus Marine domains."

*2) Combining different datasets (reanalyses and objective analyses) could be better mentioned in section 2 and 4. There are some intrinsic issues with that, in my opinion. The spread resembles mostly the offset between the products (see also point 4). The trend of the two groups of products are inherently different (Table 1), likely due to the sharp changes in the WMED around 2005-2006. For the latter, it is not easy to understand whether this is only due to a regime shift, or the Argo deployment, or a combination of them. Reanalyses seem qualitatively similar in the shift, but this can also be due to an unconstrained state before 2005. At the same time ARMOR/CORA may be constructed from a climatology that already ingests*

*Argo data, making the comparison complicated. This is beyond the scope of this work, but I recommend the authors to:*

We thank the reviewer for insightful comments. We respond to them below.

*i) mention how ARMOR/CORA are constructed (how the background is taken?)*

We added the explanation "In the CORA, the objective analysis is performed on measurement's anomalies relative to a first guess, at the 15th day of each month while in the ARMOR3D the first guess is adopted from World Ocean Atlas 2018. Both products use an objective analysis method proposed by Bretherton et al., (1976)." in the data and method section.

*ii) mention the asymmetry between the two families of products;*

We decided to add the sentence in the conclusions to underline the asymmetry as "The products with dynamically constructed ocean reanalyses and objectively analysed observations show significantly large spread at the beginning of the period of investigation while the uncertainty reduces possibly with the emergence of ARGO profilers which allowed a wider spatial and higher frequency sampling in the ocean."

*iii) be less sharp about the causes of the 2005-2006 shift.*

Thanks, we replace "which may be related to the climate regime shift in the basin (Schroeder et al., 2016) corresponding to a major deep water formation event at the beginning of the Western Mediterranean Transition (Zunino et al., 2012)" between L101-103 with "for which one of the many possible reasons is the regime shift as discussed in (Schroeder et al., 2016) corresponding to a major deep water formation event at the beginning of the Western Mediterranean Transition (Zunino et al., 2012)."

**3)** *Being the computation over the 0-300 m layer, I suggest stressing (title, abstract, conclusions) that the analysis refer to the upper ocean salinity content and anomaly, otherwise the study seems to consider the full water column*

Thanks, we agree. We made it clear in the title, abstract and conclusions. The title now reads as 'The dynamical role of upper layer salinity in the Mediterranean Sea". Considering the comment from both reviewers, we replaced "salinity content" with "upper layer salinity".

**4)** *the discussion about the spread in Section 3 should state that the spread on the content (and not that on the anomaly, not shown) is representative of the offset of the dataset, rather than their variations, and thus should be interpreted (see also point 2). This holds for both basin and local scale discussions.*

We agree with the reviewer while thinking that the offset dominates the variations possibly due to the large differences in the products, lower bound is dominated by the GREP and upper by the CORA and ARMOR3D. We added our interpretation as "The spread is representing the offset of the products more than their variability in the entire Mediterranean Sea, as well as in its eastern and western subdomains."

**Minor points / typos**

*L11 with the warming Earth -> in a warming climate*

Thanks, we changed the term.

*L12 with a large uncertainty -> with large uncertainties*

Thanks, we made it plural.

*L15 better to state in the abstract which type of products are actually used*

We added "reanalysis and in-situ objective analysis " in the abstract.

*L20 salinity content and anomaly EVOLUTION or CHANGE*

*We added "evolution". Now it reads as* "This paper investigates the salinity content and anomaly evolution in the Mediterranean Sea using observational and reanalysis products."

*L28 changes in -> changes of the essential ocean variables*

Thanks we substituted "in" with "of".

*L35 "due to strong mixing" deserves a more sound reasoning here. I suggest dropping it*

Agreed. We remove "due to strong mixing"

*L59 The mean salinity (S)*

Thanks. The mean of salinity is S with an overbar in the equation (1) so we replace it in the text accordingly.

*L75 "itself" to remove*

Thanks, removed.

*L93 also ARMOR/CORA are affected, in principle, by the the observational sampling; this sentence suggests only reanalyses are. Better to rephrase the sentence.*

We thank the reviewer and append the phrase "while the observation-based gridded products become more confident."  to the end of the sentence to appreciate the improvement in CORA/ARMOR3D.

---

## Author Comment (AC2)

**Response to the Reviewer #2 for sp-2022-11**

We thank the reviewer for the constructive feedback on our manuscript. Below are our point by point responses to the issues raised on the general aspects and minor suggestions. The reviewer comments are highlighted with italic grey typeset while our response is in normal typeset.

*Review of the Manuscript SP-2022-11 "The dynamical role of salinity content in the Mediterranean Sea" by Aydogdu et al*

*General comment to the Authors and the Editor:*

*The ms presents an analysis, based on gridded datasets of the evolution of upper layer salt content in the eastern and western Mediterranean Sea.*

*The ms is well organized, clearly written, with a logical structure that guides the reader through the author's reasoning.*

We thank the reviewer for appreciating our work.

*However it lacks a bit of a deeper and detailed discussion on what might have induced the temporal changes of water masses.*

We hope that the revisions we made throughout the manuscript following both reviewers' comments helped to improve the discussion.

*I suggest to make clear, in the title and throughout the ms, that you are referring to the upper ocean*

Thanks for the suggestion. We revised the title as "The dynamical role of the upper layer salinity in the Mediterranean Sea" and made it clear throughout the abstract and manuscript that we analyse only the upper 300 m.

*Some comparison to salinity trends from previous studies would be a valuable addition*

Thanks for the suggestion. We added a comparison and discussed it below addressing the related detailed comment of the reviewer.

*If possible, you could consider to provide spatial maps of trends, that can give an interesting perspective on where the changes are more important.*

Thanks for this interesting and relevant suggestion. Believing that it will improve the manuscript we add a figure for the upper 300 m calculated from the MEDREA24 product (longest of all) where the water column is deeper than 300 m as a fourth panel as Fig. (2d) since the space is limited to four figures in the OSRs.

[Figure]

Accordingly, we added its interpretation as "The trend calculated for each grid point from the MEDREA24, which is the analysed product covering the longest period, is presented in Fig. 2d. The dominant signal in the entire basin is positive with a larger amplitude in the Balearic Sea, Ionian Sea, Adriatic Sea, Western Levantine and with a less evident signal in the Gulf of Lions, Northern Aegean Sea and Eastern Levantine Basin. A small negative trend zone appears in the Alboran Sea." in the revised manuscript.

*I recommend publication of the ms after minor revision*

*Some more detailed comments are:*

*Title: I personally think that the title is wrong, when speaking about content it should be written "salt content" (as we would write heat content, and not temperature content), otherwise I suggest to use (not just in the title but all over the ms) "integrated salinity" (over 0-300 water column).*

We thank the reviewer and change the title as "The dynamical role of upper layer salinity in the Mediterranean Sea" following the suggestion from both reviewers.

***L10*** *I suggest to write that the Med it connected to the Black Sea, instead of the Marmara, which is only the connecting basin between Med and BS, beside being better known (wrt* Marmara) by non-Med scientists

We understand the concern of the reviewer. We replace the sentence as follows "The deficit in the water budget is balanced by the inflow in the Gibraltar Strait and Turkish Straits System connecting the Mediterranean with the less saline Atlantic Ocean and the Black Sea, respectively."

***L11*** *"The Med will be a hotspot": actually it is already a hotspot, I think this should be corrected, being a hotspots is a present status, and it implies stronger future changes that will occur.*

*We agree. We replace "will be" with "is".*

***L11-12****: I don't think the references or acronyms should go into the abstract*

We removed them as they were already redefined in the introduction.

**L27** *delete "expected to be", it is already a hotspot*

Thanks, we substituted "expected to be" with "is".

**L30** and **L31** *I would replace "water fluxes" with "freshwater fluxes"*

Thanks, done.

**L32** *it might be worthy to say what is the relative contributions in terms of freshwater of Gibraltar and Dardannels*

It is difficult to estimate precisely the relative water and salt contributions of two straits with less saline waters . We add the following information to the introduction. "...with an annual net inflow of 0.78 ± 0.05 Sv (Soto-Navarro et al., 2010) and 0.05 ± 0.04 Sv (Jarosz et al., 2013), respectively" with the references in the dedicated section.

Jarosz, E., Teague, W. J., Book, J. W., & Beşiktepe, Ş. T. (2013). Observed volume fluxes and mixing in the Dardanelles Strait. Journal of Geophysical Research: Oceans, 118(10), 5007-5021.

Soto-Navarro, J., Criado-Aldeanueva, F., García-Lafuente, J., & Sánchez-Román, A. (2010). Estimation of the Atlantic inflow through the Strait of Gibraltar from climatological and in situ data. Journal of Geophysical Research: Oceans, 115(C10).

**L32** *"which transport the less…": to be precise it is not the strait that transport the water, they allow the transport of the water…*

Thanks, we agree. We substitute the phrase "which transport the less saline Atlantic Ocean and Black Sea waters into the basin" with "from which the less saline Atlantic Ocean and Black Sea waters flow into the basin".

**L 35** and **L36** *(and everywhere else): it is now recommended to express salinity values without units, so you should remove "psu"*

We appreciate for taking our attention to this discussion on "practical salinity unit" or "practical salinity scale". We could trace back the suggestion mentioned by the reviewer back to "UNESCO (1985) The international system of units (SI) in oceanography, UNESCO Technical Papers No. 45, IAPSO Pub. Sci. No. 32, Paris, France." report. However, we were not aware of a recent discussion given that there are still many articles published using the "psu" convention. For now, we would like to keep "psu" in this contribution to the Ocean State Report 7 for consistency concerns and will be discussing it in the Copernicus Marine community.

**L41** *bigger than what?*

We add here "compared to most of the basin"

**L46** *please explain why you limit the analysis to 0-300 m, and put this clearly in the title and the abstract, since you are reporting only on the upper ocean salt content*

Thanks. We made it clear in the title, abstract and conclusions. The title now reads as 'The dynamical role of upper layer salinity in the Mediterranean Sea". Considering the comment from both reviewers, we replaced "salinity content" with "upper layer salinity".

The reason is that this work is a contribution to the Copernicus Marine Service Ocean State Report and the report seeks a consistency in the contributions from each of Copernicus Marine subdomains,the Mediterranean being one of them. The selected depth for the ocean heat content was, for example, 700 m but now Ocean Monitoring Indices are calculated in the first 300 m too. Therefore, it is a technical choice to keep all contributions from different basins coherent.

*L75 why now you are choosing 2014? This is not at all clear, and also might have some implications when you speak about the anomalies….*

We would like to emphasise that this study is a contribution to the Ocean State Report (OSR) 7 which will be proposed as an Ocean Monitoring Index (OMI) if it is published. The period to calculate the mean is chosen to be consistent with the previous contributions to OSR and current OMIs provided in the Mediterranean Sea, from the beginning of the OSR. Since the salt content anomaly OMI will be updated every year, as in example as in Fig. 4d, we would like to keep the mean that is used to calculate anomalies fixed. We note that the longer timeseries come from MEDREA24 and observational products, while GREP products are now available only to the end of 2019 even if they are regularly extended. We added a sentence as "This period is chosen to be consistent with the Ocean Monitoring Indicators produced previously in the Mediterranean Sea and other Copernicus Marine domains." with the hope that it justifies our choice.

*L76-81 maybe this could go into the Results section*

We think this calculation and the resulting plot is more a methodological content than a result that we would like to present. So we would like to keep it in the data and methods section to complement the discussion before.

*L104-105 I miss some comparison of your salinity trends with those found by previous studies*

We added a comparison for the entire Mediterranean basin with Skliris et al., (2018) in which the estimates are done from EN4 and MEDATLAS datasets between 1950-2002 in the first 150 m and 150-600 m. as "This trend is consistent with the estimates between 1950-2002 of Skliris et al., (2018) from EN4 and MEDATLAS data sets which shows a trend of 0.0096 ± 0.0077 and 0.0088 ± 0.0092 respectively, in the first 150 m while 0.0067 ± 0.0040 and 0.0067 ± 0.0036 between 150-600 m." into the paragraph presenting the discussion on the trends.

*L114 replace "the salinity from west to east" with "the upper ocean integrated salinity from west to east"*

We modified the sentence as suggested by the reviewer.

*L114-115 you use twice "such as" in the same sentence, I suggest a different wording*

Thanks for noting the repetition. We replace the phrase "such as in the North Adriatic Sea due to the presence of several freshwater inputs such as the Po river" with "such as in the North Adriatic Sea due to freshwater input from the Po river"

*L116 should be more correctly "from its low salinity" or "from its salinity minimum"*

We agree with the reviewer and replace "salinity imprint" with "low salinity".

*L130 and L131 delete "modified", the water mass you are referring to is commonly named just "Atlantic Water"*

We removed "modified" as suggested by the reviewer.

*L137 "more robust outcomes" than what?*

We intended to point out the possibility of uncertainty quantification by using more than one product but we prefer to remove the phrase "robustness" since its implication is beyond the intended meaning.

*Fig1 I suggest to use the same scale otherwise it is a bit confusing (the average basin seems to be higher in some months than the two basins' averages). Remove psu*

We thank the reviewer for the suggestion. Below we append the figure with the same scale for all. In our opinion, the variations are not clearly visible. Therefore, we prefer to keep the one in the manuscript considering also that we highlight the different y-axis scale for each curve.

[Figure]

*Fig. 2 Add the range in the caption "0-300 m"*

We added "in the upper 300 m" in the caption as suggested by the reviewer.

**Fig. 3** *I find the colourscale of (b) counterintuitive, I would rather reverse it. In the caption please use the term "integrated salinity in 0-300m"*

Thanks, we replaced Fig. 3 with the one below as suggested by the reviewer and added "integrated" in the caption.

---

## Author Response (AR2)

**Response to the Editor for sp-2022-11**

Considering the comments of the reviewers, your manuscript is returned for a minor revisions (review by the editor).

Thank you.

Please consider carefully the residual comments of reviewer 1.

We responded the reviewer 1 below as requested.

Further, I think that the differences among the various datasets should be commented more in detail. Could you add to the manuscript a plausible explanation of the different trends emerging form the various datasets?

Thank you. We add the following interpretation of ours for the differences in trends among different products. "*The differences in trend in different products that we used are mainly due to the discrepancies at the beginning of the timeseries. The weak consistency among the reanalyses visible during the first decade is likely due to the lack of observations not sufficient to constrain the different models which use different physics and initialization (e.g., Masina and Storto, 2017). The reduction of the spread among the products evolves in parallel to the increase of the observational coverage after the advent of the ARGO network. The observational products will be impacted from the scarcity of the observations in the '90s, since they rely on statistical methods.*"

Could you describe the geographical difference of the trends? Are there areas where there is substantial agreement and other areas where uncertainties are large?

In Table 1, we present the trend in the eastern and western basins separately. In Fig. 2d, we show the geographical distribution on the map for the MEDREA24 which is the closest product to the ensemble mean. We add the following interpretation to the manuscript "*Differences among different products, especially the objectively-analysed observations and GREP, for the trend is larger in the western basin. They are more confined around the mean in the eastern basin which may explain also the lower spread in this area as discussed below in Fig. (3b).*"

Please provide a revised version, that accounts for the comments of reviewer 1 and for my requests, a copy of your manuscript where all your changes are annotated and the list of your detailed answers to the reviewers' comments.

We hope that the actions taken to revise the manuscript address the suggestions from the reviewer and the editor precisely and accurately.

**Response to the Reviewer #1 for sp-2022-11**

The authors have done a good job to answer all my previous concerns, and extend the discussion to several important points (e.g. reanalyses vs objective analyses, reference period, etc.). Therefore, I am happy to recommend the manuscript for publication, beside a couple of additional minor requests.

We thank again the reviewer for kindly commenting on the revised version of our manuscript and appreciate the modifications that we did following their suggestions. Below are our point-by-point responses to the issues raised on the general aspects and minor suggestions. The reviewer comments are highlighted with italic typeset with grey fonts while our response is in default typeset.

*1) The sentence "In the CORA, the objective analysis is performed on measurement's anomalies relative to a first guess, at the 15th day of each month" is not very clear to me. Is the background also a climatology? Or a persistence? In any case, one additional suggestion for the authors is to add in Figure 2 (a-c) a climatology like WOA18, which will definitely shed light on how much OAs are close to climatology, or instead pre-Argo data were partly sufficient to sample the basin-averaged upper ocean salinity.*

We thank the reviewer for this comment and suggestion. The methodology used to produce the CORA product (INSITU_GLO_TS_OA_REP_OBSERVATIONS_013_002_b) is extensively presented in the product quality information document (QUID) listed in Table 3. Briefly, it reads as "*The first guesses are built on the basis of the monthly climatology developed by the World Ocean Atlas 2013 (www.nodc.noaa.gov/oc5/woa13). The monthly climatologies covering the period 1985- 1994,1995-2004 and 2005-2012 are interpolated to provide monthly temperature and salinity fields centered the 15 of each month. This method allows us to provide accurate first guess reproducing the climate tendencies over the covered decade.*" Therefore, the dataset is already quality controlled with respect to WOA. Therefore, with all our respect, we have to say that we find it beyond the scope of this work to assess the quality of CORA and ARMOR3D products considering the purpose of this study. To make the sentence mentioned by the reviewer clear, we substitute "*In the CORA, the objective analysis is performed on measurement's anomalies relative to a background, at the 15th day of each month while in the ARMOR3D the first guess is adopted from World Ocean Atlas 2018.*" with "*In the CORA, the objective analysis is performed on measurement's anomalies relative to a first guess provided by World Ocean Atlas 2013 monthly climatology for different decades, to accurately reproduce the climate tendencies, is interpolated and centered at the 15$^{th}$ of each month. Instead in ARMOR3D, the first guess is adopted from World Ocean Atlas 2018.*"

*2) Figure 2d: I think a more intuitive unit can be used, like psu * decade^-1*

Thanks. We replaced the Figure 2d, as suggested by the reviewer, but with psu*year^-1 to be consistent throughout the text and with earlier studies referred.

**References**

Masina S., and Storto, A., 2017. Reconstructing the recent past ocean variability: status and perspective. Journal of Marine Research, 75, (6), pp. 727-764(38).